# Intestinal Immune Deficiency and Juvenile Hormone Signaling Mediate a Metabolic Trade-off in Adult *Drosophila* Females

**DOI:** 10.3390/metabo13030340

**Published:** 2023-02-24

**Authors:** Gavriella Shianiou, Savvas Teloni, Yiorgos Apidianakis

**Affiliations:** Department of Biological Sciences, University of Cyprus, Nicosia 2109, Cyprus

**Keywords:** bioenergetics, copulation, fat body, oviposition, intestinal stem cell, leaky gut

## Abstract

A trade-off hypothesis pertains to the biased allocation of limited resources between two of the most important fitness traits, reproduction and survival to infection. This quid pro quo manifests itself within animals prioritizing their energetic needs according to genetic circuits balancing metabolism, germline activity and immune response. Key evidence supporting this hypothesis includes dipteran fecundity being compromised by systemic immunity, and female systemic immunity being compromised by mating. Here, we reveal a local trade-off taking place in the female *Drosophila* midgut upon immune challenge. Genetic manipulation of intestinal motility, permeability, regeneration and three key midgut immune pathways provides evidence of an antagonism between specific aspects of intestinal defense and fecundity. That is, juvenile hormone (JH)-controlled egg laying, lipid droplet utilization and insulin receptor expression are specifically compromised by the immune deficiency (Imd) and the dual oxidase (Duox) signaling in the midgut epithelium. Moreover, antimicrobial peptide (AMP) expression under the control of the Imd pathway is inhibited upon mating and JH signaling in the midgut. Local JH signaling is further implicated in midgut dysplasia, inducing stem cell-like clusters and gut permeability. Thus, midgut JH signaling compromises host defense to infection by reducing Imd-controlled AMP expression and by inducing dysplasia, while midgut signaling through the Imd and Duox pathways compromises JH-guided metabolism and fecundity.

## 1. Introduction

Life history theory proclaims that organismal fitness evolves by optimizing the allocation of limited resources to different energy demanding processes, which may compete and give rise to trade-offs. Reproduction and systemic host defense to infection are conserved animal fitness traits requiring energy prioritization [1]. *Drosophila melanogaster* lives and feeds on microbially contaminated matter and is thus constantly exposed to potential pathogens [2]. The *Drosophila* intestinal epithelium constitutes the first line of defense and the largest and most important host defense barrier against potential pathogens distinguishing them among indigenous gut microbiota [3]. The *Drosophila* midgut immunity includes the generation of bactericidal reactive oxygen species (ROS) by the enzyme, dual oxidase or Duox [4] and the production of antimicrobial peptides (AMPs) via the Imd and JAK/STAT signaling pathways [2,5]. However, intestinal host defense in *Drosophila* involves mechanisms beyond immunity. The midgut epithelium is protected from the external environment by a chitinous lining, the peritrophic matrix, and a thin layer of mucus [6,7]. In addition, septate junctions keep midgut epithelial cells tightly together inhibiting luminal content and microbes’ leakage into systemic circulation [3]. Furthermore, the highly acidic stomach, lysozyme excretion and gastrointestinal peristalsis, make the midgut lumen challenging for microbial colonization and infection [8,9,10]. Some pathogens, however, manage to bypass the epithelial barrier via enterocyte-damaging toxins that diffuse through or degrade the peritrophic matrix [11,12,13]. Most intestinal pathogens damage or stress the midgut epithelial barrier via toxins or metabolites and elicit the host’s immune and regenerative response [5,14]. Damaged or stressed *Drosophila* midgut enterocytes induce compensatory proliferation, a transient induction of progenitor cell proliferation and differentiation aimed to replenish the inflicted cells and re-establish epithelial barrier homeostasis [5,14,15,16]. Despite its homeostatic role, the regenerative action of progenitor cells must be controlled, because excessive cell proliferation leads to the accumulation of mis-differentiated (dysplastic) daughter cells. Chronic intestinal infection and aging share the phenomenon of progressive dysplasia that compromises *Drosophila* epithelial integrity and predisposes for tumorigenesis [14,17,18].

Immunity and epithelial regeneration are infection-induced host defenses linked to nutrient sensing pathways [19,20]. *Drosophila* female reproduction is under the endocrine control of neuropeptides, juvenile hormones (JHs) and ecdysosteroids, such as 20-hydroxyecdysone [21]. Here, we focused on JH signaling and its induction by JHs, which are sesquiterpenoid hormones primarily secreted by the *corpus allatum* (CA), a pair of endocrine glands located in *Drosophila* adults above the posterior esophagus between the brain and the ventral ganglia [22]. *Drosophila* produces three different JHs, JH III, JH bisepoxide (JHB3) and methyl farnesoate (MF), which can signal through their common JH receptors Met and Gce [23]. *JH acid methyl transferase (jhamt)* is a key gene in JHs biosynthesis and signaling. It controls JHB3 biosynthesis in the CA and the titer of JHB3 and JH III in the hemolymph during metamorphosis [23]. Apart from controlling multiple stages of larval development, JHs act as gonadotropins, controlling the production of yolk proteins in the adult fat body [21,24]. During vitellogenesis, yolk proteins (vitellogenins) are deposited in the oocytes to supply nutrients essential for embryonic development [25]. Vitellogenesis and female fecundity are also under the control of the nutrient sensing pathways target-of-rapamycin (TOR) and insulin receptor (InR) signaling [25,26].

While a trade-off between fecundity and intestinal host defense is intuitively important due to the apparent need to choose between reproduction and self-preservation, we find that such a trade-off involves only one specific aspect of the midgut defense to infection. Assessing *Drosophila* intestine motility, permeability, regeneration and three innate immunity pathways, we pinpoint a mutual antagonism specifically between Imd signaling in the midgut enterocytes and JH-mediated fecundity.

## 2. Results

### 2.1. Immune Activation Reduces Reproductive Rate

Although systemic immune response has been shown to compromise reproduction in Diptera [1,27,28], the interplay between intestinal host defense, including midgut motility, permeability and regeneration, and fecundity remains unexplored. To examine whether adult females balance fecundity with intestinal motility, which is an aspect of intestinal host defense to infection, we used three RNAi lines primarily found to control intestinal defecation rate [29]. The downregulation of *CG11307* via the VDRC lines 108230/KK and 3005/GD) and of *Myosuppressin (Ms)* via the 108760/KK line in enteroendocrine cells (EEs) using the *prospero^V1^-Gal4 (pros-G4)* driver decreased fly survival upon intestinal infection (Figure 1A). However, egg laying was also significantly decreased (Figure 1B), failing to reveal any trade-off between fecundity and intestinal motility.

To probe for the existence of a balance between intestinal permeability and fecundity, we downregulated *bbg* and *Tsp2A*, two genes necessary for septate junction formation [30,31], in progenitor cells and enterocytes, using the recombinant driver *esg-Gal4 Myo1A-Gal4 (esgMyo-G4)*. Oral infection with the virulent *Pseudomonas aeruginosa* strain PA14 [14], confirmed the significance of these septate junction genes for intestinal host defense. The downregulation of any of the two significantly decreased fly survival to infection (Figure 1C), but it did not increase egg laying; instead *bbg* downregulation significantly reduced egg laying (Figure 1D). Thus, safeguarding the intestinal barrier via septate junctions exhibits no apparent trade-off with fecundity.

A third parameter of intestinal host defense tested for its implication in a trade-off with fecundity was intestinal epithelium regeneration. The moderate inhibition of intestinal stem cell (ISC) mitosis by up to 50% upon infection is attainable by downregulating *cyclin E* (*cycE*) in the midgut ISCs using the *Delta-Gal4* driver [18]. Flies of this genotype showed accelerated fly death rate upon infection with *P*. *aeruginosa* strain PA14 (Figure 1E) without significantly altering egg laying (Figure 1F). Moreover, moderate enhancement of ISC mitosis by up to 50% upon infection is attainable by overexpressing *cycE* in the midgut ISCs using the *Delta-Gal4* driver [18]. Flies of this genotype showed a decelerated fly death rate upon infection (Figure 1E) without significantly altering egg laying (Figure 1F). Thus, mild changes in ISC mitosis able to modify host defense do not affect fecundity significantly.

Considering the known trade-off in *Drosophila* between systemic immune response and fecundity [1,32], we challenged wild-type Oregon-R females by feeding them with a solution of heat-killed bacteria before egg laying assessment. As shown in Figure 2A, feeding wild-type flies with heat-killed *P*. *aeruginosa* (strain PA14) significantly reduced egg laying, compared to unchallenged wild-type flies. Notably, a similar impact on fecundity was noticed when female flies were challenged with live pathogenic bacteria (Figure 2A). Thus, the oral stimulation of flies with live or dead bacteria compromises fecundity.

### 2.2. Imd and Duox Signaling in Midgut Enterocytes Compromises Fecundity and Insulin-Mediated Metabolism

AMP induction in the *Drosophila* midgut is attributed to the activation of the Imd and JAK/STAT pathways [5], while ROS production as part of the innate immunity response is controlled by Duox [4]. To investigate the implication of immunity pathways mediating AMP and ROS induction on fecundity, we challenged immunocompromised flies by deploying: (i) the RNAi-mediated downregulation of the Imd pathway gene *rel*, (ii) the expression of the dominant negative form of the JAK/STAT pathway receptor (*dome^ΔC^*), and (iii) the RNAi-mediated downregulation of *Duox*. Transgenes were induced in the adult midgut enterocytes using the *Myo1A-Gal4^ts^* driver by raising flies during development at 18 °C before transferring adults at 29 °C for transgene induction. Challenging wild type flies with heat-killed bacteria for 15 h was enough to compromise their fecundity (Figure 2B). The same was true with the downregulation of *dom* in the midgut enterocytes. However, the downregulation of *rel* or *Duox* in the midgut enterocytes alleviated the inhibitory effect of bacterial challenge on fecundity (Figure 2B). A similar impact of the three immune pathways on fecundity was noticed in female flies challenged with live pathogenic bacteria for 48 h. That is, virulent infection significantly compromised fecundity in wild type and *dome*-downregulated, but not in *rel-* or *Duox*-downregulated, flies (Figure 2C). Although *rel-* and *dom*-downregulated flies feeding on the vehicle medium (4% sucrose) for 48 h exhibited reduced egg laying, compared to wild type control flies, the net inhibitory effect of infection on fecundity was noticed in wild type and *dom*-downregulated flies, but not in *rel-* or *Duox*-downregulated ones (Figure 2C).

Immune response and egg production are metabolically costly processes. Enterocytes are the biggest and most numerous cells in the adult midgut, and their homeostasis requires resources. InR signaling is a major regulator of metabolism in *Drosophila* and other animals, and can be inhibited upon a systemic, fat body elicited, immune response [33]. In addition to fat body, InR signaling regulates the catabolism of energy stores in the midgut enterocytes, that is, the breakdown of enterocyte lipid droplets [34]. To assess the local impact of Imd signaling on InR, we measured *InR* expression, as well as lipid droplet mobilization, in the female midgut following the downregulation of *rel* in the midgut enterocytes and fly challenging with heat-killed bacteria. We found that the immune challenge lowered *InR* expression in the midgut (Figure 2D), and increased lipid droplets tentatively in the anterior midgut (Figure 2E) of wild type control flies. Such consequences of immune challenge were absent in *rel* RNAi females (Figure 2D,F). Therefore, the midgut enterocytes are metabolically responsive to immune challenge in a *rel*-dependent manner.

### 2.3. Mating Compromises Female Drosophila Intestinal Host Defense

While midgut immunity compromises fecundity, fecundity may also compromise aspects of midgut host defense against infection. To assess the effect of mating on intestinal host defense, we infected mated and virgin females and males via feeding with live PA14. Assessing the effect of mating on flies of four immunocompetent genetic backgrounds (Oregon-R, Berlin, *w^1118^* and *tud^1^/+*), we found that females of all genotypes die faster upon infection (Figure 3A–D), indicating that mating compromises not only systemic immunity [1], but also intestinal host defense to infection in females. However, mated and virgin males of either the Oregon-R (Figure 3E) or the *w^1118^* (Figure 3F) strain died at the same rate upon infection, indicating that mating does not compromise the intestinal host defense of males.

We then sought to investigate the aspect of intestinal host defense compromised in mated females. First, we examined intestinal motility, a host defense strategy to expunge harmful intestinal bacteria. To calculate fly defecation rate, we measured the number of excreta in wild-type Oregon-R females upon infection. Interestingly, mating did not decrease, but tentatively increased defecation rate (Figure 4A), a probable consequence of the increased food ingestion of females upon mating [35]. This tentative increase in the defecation rate upon mating could not explain but rather counteract the inhibitory effect of mating on intestinal host defense.

To assess the effect of mating on intestinal epithelium integrity, Oregon-R females were orally infected with *P*. *aeruginosa* strain PA14, and the bacterial load in the intestine and the hemolymph was evaluated by sampling and measuring colony forming units (CFUs). Mated females exhibited an increased bacterial load in their hemolymph, but not in the rest of their body, compared to virgin females (Figure 4B). To further investigate intestinal epithelium integrity, flies were fed with the unabsorbable by enterocytes blue dye, bromophenol blue, followed by the inspection of diffusion of the dye in the abdominal area of the flies. In agreement with the observed increase in bacterial load, mated flies exhibited increased blue dye diffusion (Figure 4C), confirming that mating can increase intestinal permeability. 

Intestinal barrier breakdown is closely associated with epithelial dysplasia caused by over proliferation of midgut ISCs [18]. To examine whether increased intestinal permeability upon infection in mated females corelates with excessive ISC proliferation, wild-type Oregon-R virgin and mated flies were orally infected with *P*. *aeruginosa* strain PA14 and the number of midgut cells undergoing mitosis was counted. Indeed, mating increases the number of dividing cells, as revealed by phospho-Histone H3 staining (Figure 4D). To verify if ISC over-proliferation due to mating leads to epithelial dysplasia over time, female progeny of Oregon-R flies crossed to *Delta-Gal4 UAS-gfp* were assessed for clusters of ISC-like cells and EEs. A significant increase in ISC-like cell clusters was observed in 4-day-old, mated females compared to virgin females of the same age, and this difference was more pronounced in 30-day-old females, which exhibited more clusters and more cells per cluster (Figure 4E). Moreover, mated 30-day-old females also exhibited an increase in the clusters of EEs (Figure 4F), suggesting a breach in epithelial cell homeostasis. 

The *Drosophila* intestine is armed against infection with midgut enterocyte produced AMPs, such as *attA* and *diptB*, which are induced by the Imd pathway, and *drsl3*, induced via the JAK/STAT pathway [5]. To explore the effect of mating on intestinal AMP induction upon feeding with heat-killed bacteria, we measured the expression of *attA, diptB, drsl3* and *Duox* in the midgut and the body wall carcass, a systemic set of tissues. We found that only *attA* and *diptB* are transcriptionally inducible upon feeding with heat-killed bacteria. Importantly, mating significantly reduced the systemic induction of *attA* and *diptB* expression upon bacterial challenge (Figure 4G). On the other hand, mating significantly reduced the baseline (rather than the induction) of *diptB* expression (and tentatively of *attA* expression) in the midgut upon bacterial challenge (Figure 4H). Thus, mating inhibits the systemic response as well as the local, midgut expression of Imd pathway-controlled AMPs.

### 2.4. JH Synthesis and Signaling Compromises Intestinal Host Defense

To further assess the cost of fecundity on intestinal host defense, we manipulated JH signaling genetically. Egg laying in *Drosophila* relies on the synthesis of yolk proteins in the fat body and their uptake by the ovaries, and it is controlled by the production and release of JHs from CA [22]. To assess the systemic role of JH on intestinal host defense, we downregulated *jhamt,* an enzyme-coding gene required for the latest steps of JH biosynthesis. The downregulation of *jhamt* in the CA using the *Aug21-Gal4* crossed with either of the two *UAS-jhamt^RNAi^* lines tested reduced egg laying (Figure 5A), confirming the requirement for JH synthesis in CA for normal fecundity. The downregulation of *jhamt* also resulted in flies surviving longer upon infection with live PA14 (Figure 5B), suggesting that systemic JH synthesis promotes egg laying and compromises host defense to intestinal infection.

Taking into consideration the secretory role of the midgut EEs, we downregulated *jhamt* using the EE-specific driver *pros-Gal4*. The downregulation of *jhamt* decreased egg production and increased survival to infection (Figure 5C,D). JH is also produced in the progenitor cells of the adult *Drosophila* midgut and acts in an autocrine manner facilitating ISC mitosis and the regeneration in wild type flies [36]. To investigate whether its synthesis in progenitor cells partakes in a trade-off between fecundity and intestinal host defense, we downregulated *jhamt* using the progenitor-specific driver *esg-Gal4^ts^*. *jhamt* downregulation in progenitors tentatively reduced fecundity, and increased survival to infection (Figure 5E,F). On the other hand, the downregulation of *jhamt* in adult midgut enterocytes using *Myo1A-Gal4^ts^* driver, did not significantly affect egg laying or survival to infection (Figure 5G,H). We conclude that systemic and local JH production in EEs and possibly in progenitor cells contributes to fecundity and counteracts host defense to infection.

JH is known to induce yolk protein production in the fat body [22]. Accordingly, we downregulated yolk protein genes, *Yp1, Yp2* and *Yp3*, and the cytoplasmic JH receptor genes, *Met* and *gce*, and their coactivator gene, *tai*, using the *Cg-Gal4* driver, which is expressed in the fat body and hemocytes. Egg laying was significantly reduced upon the downregulation of *Yp1, Yp2* and *Met* (Figure 6A). Moreover, the downregulation of JH receptor genes, *Met* and *gce*, and coactivator, *tai,* resulted in a significant increase in survival to intestinal infection, while the downregulation of yolk protein genes exerted a marginal but significant increase of host defense (Figure 6B). We conclude that JH signaling and yolk protein gene expression in the fat body and hemocytes contribute to fecundity and counteract host defense to infection.

To assess the autocrine effect of progenitor-produced JH on intestinal host defense to infection and fecundity, we downregulated *Met*, *gce* and *tai* using the *esg-Gal4^ts^* driver. We noticed a significant decrease in fecundity upon the downregulation of *Met*, *gce* and *tai* (Figure 6C), and a significant increase in survival to infection upon the downregulation of *gce* and *tai* (Figure 6D). Thus, not only JH production, but also JH signaling reception, in the midgut progenitors promotes a trade-off boosting high fecundity, while compromising intestinal host defense.

To pinpoint the aspect of host defense compromised by JH signaling in the midgut progenitors, we downregulated *gce* using the *esg-Gal4^ts^* driver. In agreement with a previous study [37], JH signaling was required for high ISC proliferation. A mitosis assessment showed that the number of dividing cells at baseline and upon intestinal infection with *P*. *aeruginosa* strain PA14 was radically lower in *gce* RNAi flies (Figure 7A). This suggests that JH signaling may contribute to high proliferation, which is linked to higher host defense to intestinal infection, but also to progenitor cell mis-differentiation upon infection that reduces survival [18]. Accordingly, we noticed that progenitor cell clustering was reduced upon *gce* downregulation during aging (Figure 7B). Furthermore, high progenitor cell proliferation and clustering may compromise intestinal epithelium integrity. Accordingly, we noticed that downregulation of *gce* in progenitors decreased the fraction of flies exhibiting dye diffusion in their abdominal cavity upon infection (Figure 7C).

We further examined whether JH signaling in the progenitor cells affects intestinal innate immune signaling. We used *gce* RNAi and control flies, which were either untreated or challenged with heat-killed bacteria, and measured the adult midgut expression of *attA, diptB, drsl3* and *Duox* genes. We found that the downregulation of *gce* significantly increased the expression levels of *diptB* in untreated and bacterially challenged flies, suggesting that JH signaling in progenitors inhibits the baseline expression of *diptB* (Figure 7D).

Finally, to assess if JH signaling in the midgut progenitor cells affects metabolic signaling, we assessed *InR* expression upon the downregulation of *gce* using the *esg-Gal4^ts^* driver. We noticed that *gce* downregulation lowered the baseline of *InR* expression in the midgut (Figure 7E), suggesting that JH signaling operates in the midgut progenitors according to a trade-off that facilitates InR signaling and fecundity, but compromises intestinal host defense by increasing dysplasia and lowering *diptB* baseline expression. In support of our conclusions, we noticed that *tud^1^* homozygous mothers producing sterile *tud^1^/+* offspring exhibited pleiotropic defects regardless of their virgin vs mated status, including severely compromised AMP expression (Figure 8E) and stem cell mitosis (Figure 8B), as well as excessive dysplasia (Figure 8C) and leakiness of the gut (Figure 8D). Compared to their isogenic fertile counterparts produced by *tud^1^/+* mothers, the sterile flies were susceptible to infection regardless of their virgin vs mated status (Figure 8A), indicating that virginity benefits survival to intestinal infection only if AMP expression and cell homeostasis in the midgut is functional.

## 3. Discussion

Studies in insects suggest that fecundity inhibits the activation of systemic immune response [1]. However, the link between fecundity and host defense cannot be generalized. Comparing fecundity and survival to intestinal infection in 143 fly lines of the Drosophila Genetic Reference Panel (DGRP), we found no correlation between the two traits [38]. This is probably because host defense is multifaceted and includes a systemic and a local tissue component. The systemic defense to infection is physiologically different from the intestinal one in terms of innate immune response. The former relies on toll and Imd pathway induction and the release of AMPs from the fat body and hemocytes into the hemolymph, while intestinal immunity requires Imd and JAK/STAT pathway signaling in enterocytes to produce luminally secreted AMPs [5,39]. Moreover, intestinal epithelium integrity and regeneration and intestinal motility are parameters of host defense directly relevant to intestinal infection [18,29]. Intestinal motility inhibition by downregulating *CG11307* and *Myosuppressin* in EEs decreased host defense to intestinal infection, but also female reproductive output. *CG11307* and *Ms* are lowly expressed in the midgut EEs compared to other gut motility regulators [40,41] but their downregulation in the midgut EEs effectively reduces the defecation rate [29]. Thus, *CG11307* and *Ms* are *bona fide* intestinal motility regulators contributing to the intestinal host defense to infection, without counterbalancing fecundity.

Similarly, compromising intestinal epithelium integrity by downregulating *bbg* and *Tsp2A*, two genes encoding septate junction proteins, in the adult midgut progenitors and enterocytes decreased host defense to intestinal infection, but did not increase fecundity in the absence of an infection. To the contrary, we noticed a mild reduction in egg laying upon *bbg* downregulation, which could result from the malabsorption of nutrients by leaky midgut cell-cell adhesion. Moreover, a moderate change in the rate of intestinal epithelium regeneration either via overexpression or via downregulation of *cycE* in the midgut stem cells changed host defense to intestinal infection, but it did not modulate fecundity in the absence of an infection. Thus, intestinal motility, integrity and regeneration partake to the intestinal host defense without compromising fecundity in the absence of a bacterial infection. However, the induction of innate immune signaling by heat-killed bacteria decreased egg laying. Not only heat-killed bacteria but also live pathogenic bacteria compromised fecundity through innate immune signaling. The downregulation of the Imd or Duox pathway (but not of the JAK/STAT pathway) alleviated the cost to fecundity upon heat-killed or live bacteria challenge. Therefore, Imd and Duox pathway induction in the midgut enterocytes may negatively regulate fecundity.

If intestinal immune defense and egg production are both metabolically costly, Imd signaling in midgut enterocytes may regulate metabolic pathways and in turn fecundity. Accordingly, we find that immune activation with heat-killed bacteria lowers *InR* expression in the midgut and increases lipid droplet accumulation in the female midgut enterocytes of wild type control flies, but not of *rel* RNAi flies. This suggests that Imd signaling compromises fecundity by lowering InR signaling and midgut lipid utilization; and agrees with data showing that *rel* inhibits metabolic signaling and diet-dependent microbial abundance in the fly intestine [19].

Of note, feeding flies with heat-killed bacteria was not lethal to flies for the duration of our experiments. Moreover, prolonged feeding of flies with highly immunogenic, but very low in virulence strains of *P*. *aeruginosa*, such as the strain CF5 [42], induces antimicrobial peptide expression in the intestine, without killing almost any flies for at least 20 days at 29 °C [43,44]. Thus, intestinal bacteria associated molecular patterns seem unable to induce an immune burden to flies. However, bacterial infection in the brain and prolonged induction of innate immunity causes neurodegeneration and reduces longevity mediated by the Imd pathway [45]. Other fly tissues though may not respond to innate immunity the same way, because, for example, the targeted inhibition of *rel* in the fly brain increases longevity, while the ubiquitous inhibition of *rel* shortens longevity [45]. Accordingly, brain protection against infection, seems to operate in a trade-off with longevity. By responding to brain bacteria via *rel*, flies are coping with the microbial challenge, but they exhibit early aging in the form of neurodegeneration and reduced locomotion [46].

Interestingly, mated flies exhibit reduced resistance to intestinal infection. Previous studies have established that mating remodels the female midgut in anticipation of nutritional needs, and that the male sex peptide received by females upon copulation controls almost all of the midgut transcriptional changes attributed to mating, including the higher expression of protein and lipid metabolism genes, which may support reproduction [37,47]. Here we undertook a holistic approach to assess intestinal host defense against a specific pathogen in comparing virgin vs. mated flies by measuring, not only fly survival upon mating, but also four additional aspects of intestinal host defense, namely, intestinal motility, intestinal permeability via CFU and Smurf assessment, intestinal regeneration and dysplasia, and systemic and local antimicrobial peptide expression. We pinpoint two aspects of intestinal host defense compromised by mating in adult flies. One pertains to the increase of midgut compensatory cell proliferation upon infection due to mating. While midgut regeneration helps when coping with an intestinal infection, excessive stem cell proliferation leads to the loss of proper cell differentiation, accumulation of dysplastic cells, epithelium integrity disruption, and thus compromised host defense [14,18]. Such dysplastic phenotypes are also observed in the midguts of old flies [17,18,48]. We now show that mating leads to midgut dysplasia typified by excessive stem cell proliferation and the accumulation of mis-differentiated cells upon aging. Dysplasia renders midgut epithelium integrity susceptible to infection, which is evident by the increased diffusion of luminal bromophenol blue into the abdominal cavity of mated flies, and the increased bacterial load in the mated fly hemolymph. Considering that the increased progenitor cell mitosis observed upon mating remodels the midgut epithelium to accommodate reproductive output [37], the increase in mitosis and its diversion towards dysplasia upon infection in mated female midguts is the combined outcome of midgut remodeling due to mating and the damage imposed to enterocytes due to infection.

In addition to the known effect of mating on systemic AMP expression [32,49], we noticed a significant decrease in *diptB* expression in the midguts of mated flies, suggesting that mating compromises intestinal Imd signaling as well [5]. JAK/STAT signaling pathway, on the other hand, is not compromised upon mating, as shown by the expression levels of its target gene *drsl3*. We suggest that the excessive ISC proliferation, accumulation of dysplastic cells and concomitant midgut epithelium integrity loss, as well as Imd pathway suppression, cause the decrease in survival to infection due to mating. On the other hand, intestinal motility is not compromised upon mating, but instead, a tentative increase in defecation rate was observed. The increase observed in motility is most likely due to the increased feeding behavior of females upon mating aiming to fulfill their energetic demands [37,50]. However, the higher defecation rate does not suffice to decrease intestinal bacterial load, probably because it is compensated by the higher ingestion of fly food and the bacteria it contains.

To further investigate the cost of fecundity on intestinal host defense, we manipulated fecundity genetically, considering that reproductive output in Diptera relies heavily on the synthesis of yolk proteins in the fat body and their subsequent uptake by the ovaries. JHs are secreted upon mating from the CA of inseminated *Drosophila* females in response to sex peptide, which is found in the seminal fluid and regulates yolk protein synthesis and uptake by oocytes, but it can also suppress systemic innate immunity [32]. Here, we confirm the systemic role of JH in the regulation of egg production rate and the suppression of humoral immune defense by genetically reducing *jhamt* expression in the CA. Moreover, JH signaling inhibition in the fat body and hemocytes via the downregulation of the JH receptors, *Met* and *gce,* and their co-activator *tai*, as well as the downregulation of *Yp2* and *Yp3,* increased survival to intestinal infection. Notably, the downregulation of *Met, gce* and *tai* increased survival more prominently compared to the downregulation of individual yolk protein genes, probably due to the functional redundancy of the latter. The results, however, suggest that JH signaling and yolk protein production in the fat body compromises intestinal host defense.

JHs are synthesized in the CA but also in the adult midgut progenitors [36]. In this study, we sought to investigate whether the synthesis of JHs in the midgut cells operates in the trade-off between fecundity and intestinal host defense. We noted an increase in survival upon infection with *P*. *aeruginosa* and a decline in the egg laying rate when *jhamt* was downregulated in either progenitor cells or EEs. On the other hand, midgut enterocyte-derived JHs did not affect fecundity or survival to infection. The downregulation of genes encoding JHs receptors, *Met* and *gce,* and their coactivator *tai,* in the progenitor cells resulted in significantly increased survival to infection, and decreased egg laying. The effects on survival and fecundity were more obvious upon the downregulation of *gce*, suggesting that JHs acts mainly through Gce in the midgut progenitors rather than the alternative JHs receptor, Met. Noticeably, the downregulation of *gce* in progenitors caused more prominent changes in the egg laying capacity and the intestinal host defense compared to the downregulation of JHs biosynthesis genes in the progenitors or the EEs, suggesting that JHs signaling in the midgut relies on locally as well as systemically produced JHs.

To pinpoint the aspect of host defense compromised upon JH signaling in intestinal progenitor cells, we examined whether ISCs regeneration and dysplasia are affected upon the downregulation of *gce*. We observed a significant decrease in mitosis and dysplastic cell accumulation, and reduced permeability in the midguts of *gce^RNAi^* females. While JHs facilitate ISCs proliferation, midgut remodeling and in turn reproductive output [37], we suggest that excessive ISCs proliferation upon infection causes dysplasia and disrupts midgut epithelium integrity. This agrees with the finding that Apc-Ras-induced tumors require *gce* expression to grow and survive in the midgut [36].

We also tested whether AMPs generation by innate immune signaling is affected by JH signaling in the gut. Interestingly, we observed that *diptB* expression is increased upon the downregulation of *gce* in the midgut. Similarly, *diptB* expression is reduced in females upon mating, presumably due to the increased JH signaling in flies upon mating. JH signaling in the gut may contribute to nutrient delivery from the intestine towards reproduction, since JH regulates lipid metabolism in the midgut enterocytes [37]. Accordingly, we show that *InR* expression is lower upon *gce* downregulation in the midgut progenitors. Thus, mating and JH production may negatively impact the intestinal immune response in addition to the systemic one via InR signaling [32].

In summary, our findings suggest that the trade-off between fecundity and immunity takes place systemically, but also locally in the female midgut allocating resources between fecundity and intestinal host defense (Figure 9). We suggest that the synthesis of JHs in the CA and the midgut progenitors and EEs contribute, in a hormonal, autocrine and paracrine manner, to the proliferation of midgut ISCs and the local and systemic suppression of AMP expression. Future work may clarify the cell and tissue specificity of JHs biosynthesis enzymes in the production of different types of JHs. Biochemical analysis may, nevertheless, be complex, because midgut enteroendocrine or progenitor cells may secrete JHs mainly in a paracrine or endocrine way and may not contribute significantly to the pool of JHs present in the hemolymph.

## 4. Materials and Methods

*Drosophila* maintenance. Fly stocks were reared on a standard agar/cornmeal diet (1% Agar, 3% Yeast, 5% Sugar, 6% cornmeal, 2.56% Tegosept and 0.38% propionic acid) in vials with ~5 mL of fly food at 18 °C and 65% humidity, on a 12:12 light:dark cycle. Strain cultures and crosses were set up at 18 °C in bottles containing 50 mL of fly food. Male and female offspring was collected at 20 days and allowed to mate (mated) or sexes were kept separately (virgins) at 18 °C for 3 days. Then they were transferred at 25 °C, except for those containing tub-Gal80ts, which were transferred to 29 °C for 5 days to allow the expression of UAS transgenes. Directly compared genotypes and treatments were handled in parallel.

*Drosophila* strains. Progenitor expression: *w; esg-Gal4 UAS-GFP tub-Gal80^ts^*. ISC expression: *w; UAS-src-GFP/CyO; Delta-Gal4 tub-Gal80^ts^/TM6C (Dl-G4 U-GFP)* or *w; esg-Gal4 UAS-GFP; Su(H)-Gal80 tub-Gal80^ts^* (ISC^ts^). EC expression: *tub-Gal80^ts^/FM7; Myo1A-Gal4 UAS-EGFP/CyO.* EC and progenitor expression: *w; esg-Gal4 UAS-GFP Myo1A-Gal4* (chromosome II). EE expression: *w; pros^V1^-Gal4/TM6C*. For fat body and hemocyte expression: *w; Cg-Gal4* (BDSC#7011). CA expression: *w; Aug21-Gal4/CyO* (BDSC#30137). UAS lines for jhamt^RNAi^ (103958 and 19172), Met^RNAi^ (100638 and 10801), gce^RNAi^ (11176), tai^RNAi^ (15709), bbg^RNAi^ (15975/GD), Myosuppressin^RNAi^ (108760/KK), CG11307^RNAi^ (108230/KK and 30050/GD), cycE^RNAi^ (110204/KK), and dDuox^RNAi^ (330085) were obtained from VDRC. UAS lines of Yp1^RNAi^ (67219), Yp2^RNAi^ (55931), Yp3^RNAi^ (67220), Tsp2A^RNAi^ (40899), rel^RNAi^ (33661), *w; UAS-CycE* (4781), and *w; UAS-Dome^ΔCYT3.2^* were obtained from Bloomington Stock Center. *tud^1^ sp^1^ bw^1^ /CyO* and Berlin lines were gifts from Michelle Arbeitman. *w^1118^* was used in crosses as a control to UAS strains and Oregon-R, *w^1118^* and Berlin as pure strains.

Fly infection with live *Pseudomonas aeruginosa*. A 3 mL overnight culture of *P*. *aeruginosa* strain PA14 [51] grown in Lysogeny Broth (LB) was diluted 1:100 in LB to prepare a 30 mL overday culture of OD_600nm_ = 3. For 5 mL of infection mix, 3.5 mL ddH_2_O, 1 mL 20% sucrose and 0.5 mL PA14 of OD_600nm_ = 3 were mixed. For 5 mL of the control mix, 4 mL ddH_2_O were mixed with 1 mL 20% sucrose. Then, 5 mL of the infection or control mix was added on a compressed cotton ball at the bottom of a narrow fly vial, and a dry cotton ball plugged the top. After 4–5 h of starvation in empty vials, the flies were put in the infection vials and incubated at 25 °C upside down for 48 h. The percentage of dead flies was calculated daily and the LT50% (lethal time 50%), the time required for 50% of the flies to die indicated the average fly survival.

Defecation rate assessment. Flies were fed with a concentrated bacterial paste of OD_600nm_ = 50, prepared from an overday culture of *P*. *aeruginosa* of OD_600nm_ = 2, that was concentrated 25 times by centrifuging the culture and dissolving the pellet in 4% sucrose. Each feeding vial plugged with a cotton ball contained 5 mL of agar gel (3% *w*/*v* agar in H_2_O) covered with a Whatmann disc, which was then soaked with 200 μL of the infection mix. In total, 25 mated starved young female flies were allowed to feed for 15 h at 25 °C on the infection mix (or 4% sucrose for control) in each vial. To visualize excretions, flies were fed with bromophenol blue solution for 5 h, then were transferred for 24 h in sterilized petri dishes containing sterilized cotton ball halves impregnated with 2.5 mL of the bromophenol blue solution. Excreta per fly per day were enumerated from three replicates per condition.

Fly feeding with heat-killed *P*. *aeruginosa*. A bacterial culture of OD_600nm_ = 50 was prepared in 4% sucrose (as above) and heat-killed at 70 °C for 5 min. The flies were transferred for 15 h in sterile vials containing 2 mL of 4% sucrose 2% agar covered with fitting Whatman discs, which were then soaked with 200 μL of the heat-killed bacterial culture.

Smurf assay. Flies were first fed for 4 days with a fly food containing 50 μg/mL rifampicin (Sigma, St. Louis, MO, USA). Then flies were allowed to feed on a concentrated PA14 culture of OD_600nm_ = 50 for 15 h, starved in sterile polystyrene vials for 5 h, and finally transferred for another 5 h in sterile polystyrene vials each containing a cotton ball impregnated with 5 mL bromophenol blue solution (0.5% of the dye powder dissolved in 4% sucrose and adjusted to pH = 7). The inspection of the level of diffusion of the blue dye in the abdominal area of the flies followed, based on which the flies were categorized as Smurfs (dye diffusion outside the gut) vs. non-Smurfs (dye retention in the gut).

Bacterial load. Flies were first fed for 4 days with fly food containing 50 μg/mL Rifampicin. Then, flies were let feeding on a concentrated PA14 culture of OD_600nm_ = 3 for 48 h (infection mix contained 4 mL of the bacterial culture and 1 mL 20% sucrose). Then flies were externally sterilized, by briefly dipping them into pure ethanol, and dissected in a droplet of 50μL of PBS (phosphate-buffered saline), separating the thorax from the abdomen and letting the hemolymph diffuse in the PBS droplet. Then, 50 μL of the hemolymph samples were spread on LB agar plates containing rifampicin (50 μg/mL). Dissected flies were first placed in tubes containing 0.8 mL PBS and a stainless-steel bead of 5 mm diameter (Qiagen, Germantown, MD, USA) and were homogenized using the TissueLyser II (Qiagen) at 50 Hz for 5 min, diluted (1:10,000) and 15 μL were spread on LB plates, and incubated at 37 °C for 16 h. CFUs (colony forming units) from three replicates per condition were enumerated.

Fecundity Assessment. Approximately 45 females were mated to 15 males in bottles with 50 mL fresh fly food for four days. Then, the females were transferred to empty bottles covered with sterilized petri dishes (35 mm × 10 mm) containing 4 mL pasteurized multi-fruit juice, 2% agar, 2.56% Tegosept and 0.38% propionic acid, supplemented in the middle with a scoop of heat-killed yeast paste (2 g of heat-killed yeast in 3 mL ddH_2_O). Seven to eight females were transferred for 4 days every 12 h to new bottles covered with fresh fruit juice plates where they laid their eggs. The mean fecundity was calculated based on the number of eggs laid per female fly per day.

Generation of germline-less flies. Homozygous *tud^1^* mothers were crossed to wild-type Berlin males to produce germline-less *tud^1^/+* offspring. Genetically identical controls having a germline were generated by crossing Cyo/*tud^1^* mothers to wild-type Berlin males, selecting the *tud^1^/+* offspring.

Immunostaining. Flies were starved for 5 h and 10 or more midguts per genotype were dissected on a Sylgard-coated petri-dish in 1X PBS (isotonic solution) and fixed in 4% formaldehyde for 30 min at room temperature. This was followed by blocking in PBT (1X PBS, 0.2% Triton, 0.5% BSA) for at least 20 min at room temperature and overnight incubation at 4 °C with primary antibodies: rabbit-anti-pH3 (Millipore 1:4000, Burlington, MA, USA), mouse-anti-prospero (DSHB 1:100), chicken-anti-GFP (Invitrogen 1:1000, Waltham, MA, USA). Then, 1–2 h of secondary antibody incubation was carried out at room temperature in the dark: mouse, rabbit or chicken were conjugated to Alexa fluor 488 and 555 (Invitrogen 1:1000) and DAPI (Sigma, 1:3000 of 10 mg/mL stock) to visualize nuclei. Tissues were mounted on glass microscope slides with 20μL Vectashield (Vector, Burlingame, CA, USA) and visualized under fluorescent microscope (Zeis Axioscope A.1, Zeis, Oberkochen, Germany) at 20× magnification to count the number of pH3 cells or at 40× magnification to count the number of *Dl-Gal4 UAS-GFP* and prospero positive cell clusters along the whole midgut.

RT-qPCR. For each genotype, triplicates of 20–25 midguts were dissected on dry ice and stored at −80 °C. RNA extraction and cDNA synthesis using the TaKaRa Prime ScriptTM RT Master Mix Kit (TaKaRa, Kusatsu, Japan) followed. qPCR amplification was performed using the gene specific primers shown below and the following amplification program: 95 °C for 30 s (initial denaturation), 40 cycles of 95 °C for 10 s (denaturation), 60 °C for 30 s (annealing), 65 °C for 30 s (extension) and 65 °C for 1 min (final extension). The expression levels of the genes of interest were normalized to the expression levels of two reference genes, *rpl32* and *a-tub*, using the 2-ΔΔCt method. Data were analyzed using the Bio-rad CFX Manager 3.1 program (Bio-rad, Hercules, CA, USA).

*a-tub*: L: GCTGTTCCACCCCGAGCAGCTGATC;

   R: GGCGAACTCCAGCTTGGACTTCTTGC

*rpl32*: L: CGGATCGATATGCTAAGCTGT; R: CGACGCACTCTGTTGTCG

*attA*: L: CCTTGACGCACAGCAACTTC; R: CCGATCCCGTGAGATCCAAG

*diptB*: L: ATCTGCAGCCTGAACCACTG; R: AAGGTGCTGGGCATACGATC

*drsl3*: L: GGAGGCCAACACTGTTTTGG; R: CAGCAAGGACCTCCGAAAGT

*Duox*: L: CACGCGCAGCAGGATGTAAGGTTT;

   R: GCTGCACGCCAACCACAAGAGACT

Statistical analysis. For the fly survival assessment analysis, the Kaplan–Meier method was applied using the log-rank test (MedCalc statistical software). A two-tailed *t*-test was applied to analyze pH3 counts for the midgut regeneration assessment. For fecundity, bacterial load and defecation rate assessment the Mann–Whitney U test was applied. Fisher’s exact test was applied for the Smurf assay analysis. An ANOVA single-factor test was applied for RT-qPCR analysis. A *p*-value < 0.05 is labeled as “*”, while *p*-values < 0.01 and <0.001 are labeled as “**” and “***”, respectively. “ns” stands for not statistically significant. Error bars represent the standard deviation of the mean (STDEV).

## Figures and Tables

**Figure 1 metabolites-13-00340-f001:**
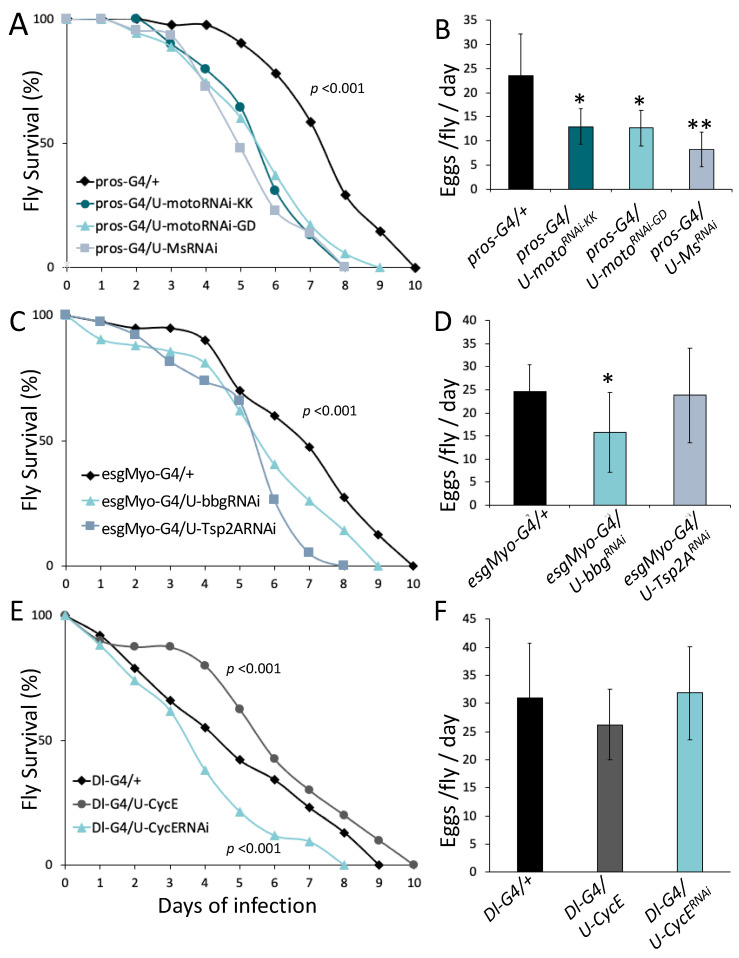
Fecundity does not increase when midgut motility, progenitor and enterocyte septate junctions, or regeneration is compromised. Downregulation of the motility-related genes *moto* (*CG11307* gene lines 1082330/KK and 30050/GD, LT50 = 5.5; *p* < 0.0001) and *Myosuppressin* (*Ms*) (LT50 = 4.9; *p* < 0.0001) resulted in a significant decrease in survival upon infection, relative to control flies (LT50 = 7.3) (**A**), but also caused a significant reduction in eggs per fly per day (*p* < 0.05) (**B**). Downregulation of the genes encoding septate junction proteins *bbg* (LT50 = 5.5; *p* < 0.05) and *tsp2A* (LT50 = 5.4; *p* = 0.0001) resulted in a significant decrease in survival upon infection, relative to control flies (LT50 = 6.8) (**C**). *bbg^RNAi^* also caused a significant reduction in eggs per fly per day (*p* < 0.05) (**D**). Downregulation and overexpression of *cycE* in ISCs (LT50 = 4.5; *p* < 0.05) resulted in a decrease and an increase in survival upon infection, respectively, relative to control (LT50 = 5.3) (**E**) and had no effect on fecundity (**F**). *n* = 45 flies for (**A**,**C**,**E**). Six replicates of 7–8 flies each for (**B**,**D**,**F**); “*” and “**” stand for *p*-value < 0.05 and <0.01, respectively.

**Figure 2 metabolites-13-00340-f002:**
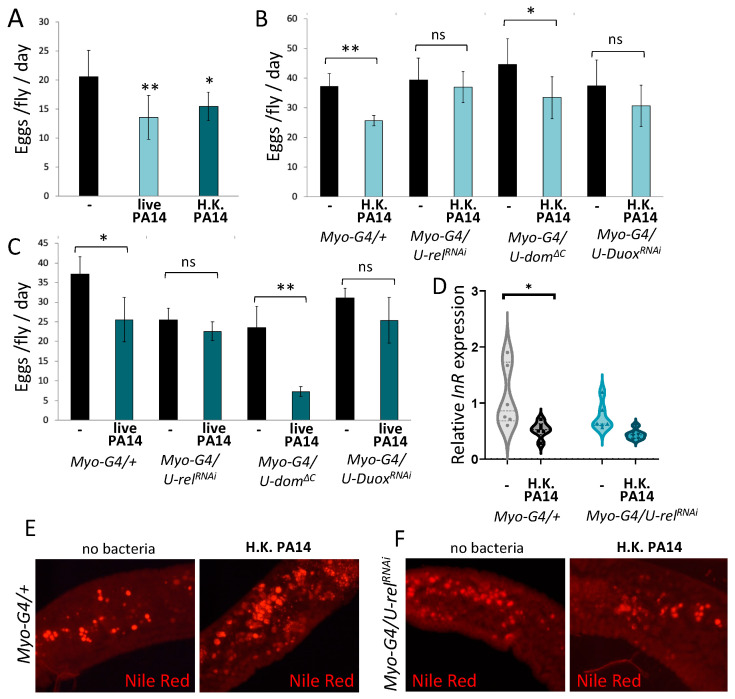
Fecundity and enterocyte metabolism are compromised by the midgut innate immunity. (**A**) Wild-type *Oregon-R* females fed with live (*p* < 0.01) or heat-killed (*p* < 0.05) *P*. *aeruginosa* PA14 reduced egg production per fly per day, compared to mock-infected flies. (**B**) Feeding with heat-killed bacteria in control flies reduced egg laying (*p* < 0.01), but not upon downregulation of *rel* or *Duox* in enterocytes via *MyoG4^ts^*. Eggs per fly per day was significantly reduced upon JAK/STAT pathway inactivation through the expression of *dome^ΔC^* (*p* < 0.05). (**C**) Feeding with live PA14 caused a significant reduction in egg laying in control flies (*p* < 0.05). The reduction was rescued upon downregulation of *rel* and *duox*, but not upon expression of *dome^ΔC^* (*p* < 0.01) in the midgut enterocytes. (**D**) Immune challenge with heat-killed PA14 lowered *InR* mRNA levels in *Myo-G4/+* females, as measured by RT-qPCR (*p* < 0.05), and *InR* mRNA levels remained low in *Myo-G4/U-rel^RNAi^* females, regardless of immune challenge. (**E**) Nile Red stained lipid droplets increased in the anterior midgut of *Myo-G4/+* females upon immune activation with heat-killed PA14. (**F**) Lipid droplets in *Myo-G4/U-rel^RNAi^* females did not increase upon immune challenge. Representative images for each condition are shown (**E**,**F**). “ns”, “*” and “**” stand for *p*-value > 0.05, <0.05 and <0.01, respectively.

**Figure 3 metabolites-13-00340-f003:**
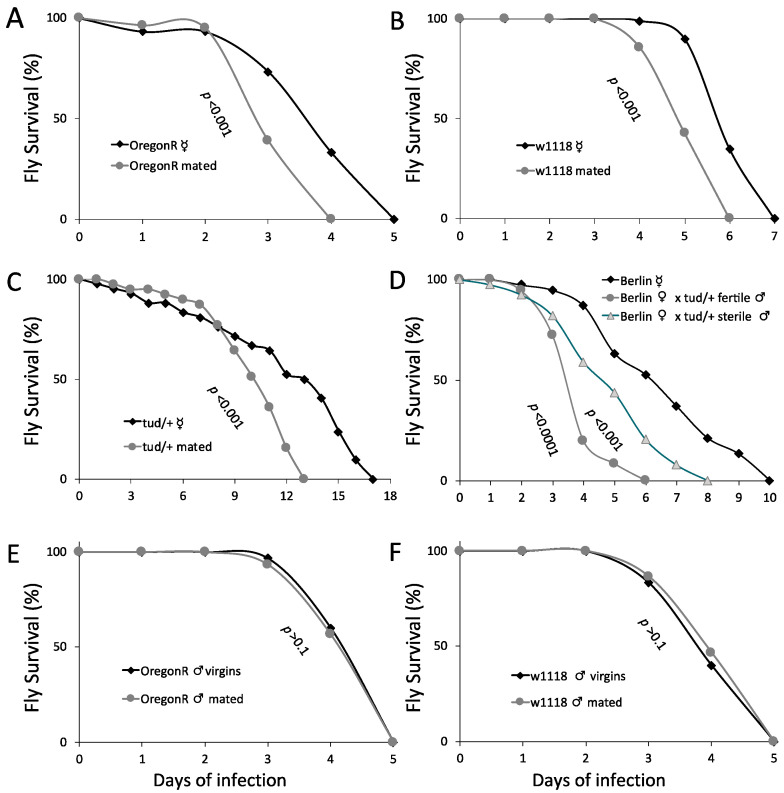
Mating compromises host defense to intestinal infection in immunocompetent females of various genetical backgrounds. Mating significantly decreased survival to infection with *P*. *aeruginosa* PA14 in Oregon-R (**A**), *w^1118^* (**B**), *tud^1^/+* (**C**) and Berlin (**D**) females (*n* = 45). Mating had no effect on Oregon-R (**E**) and *w^1118^* (**F**) males (*n* = 30).

**Figure 4 metabolites-13-00340-f004:**
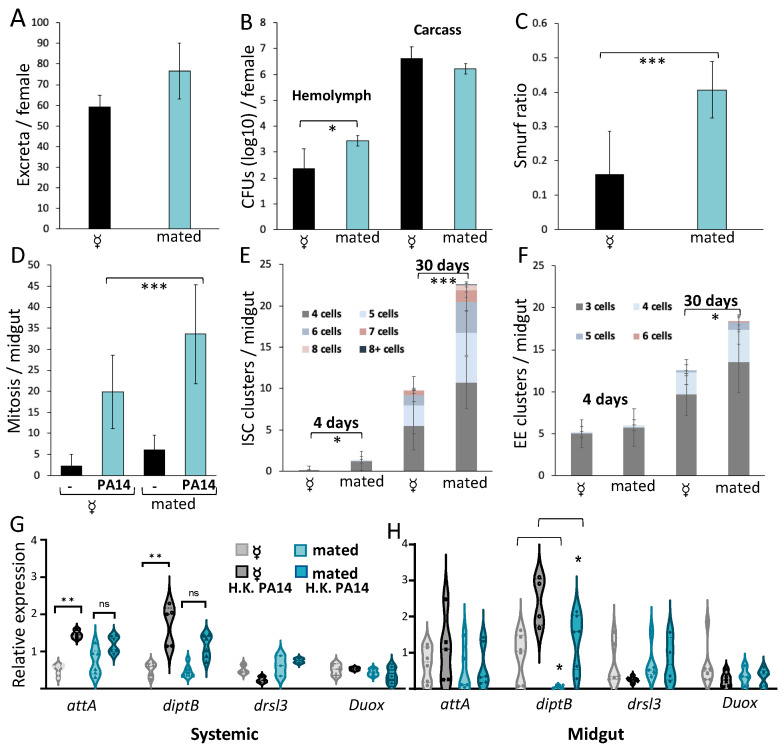
Mating increases intestinal permeability, aging-related epithelial dysplasia and suppresses local innate immune signaling. (**A**) Average and standard deviation of excreta per female per day from 3 replicates of 10 flies. The defecation rate of Oregon-R mated flies was not compromised compared to virgin females. (**B**) Average number of CFUs in the hemolymph and fly carcass from 6 replicates of three females. (**C**) Ratio of Smurf to total flies from 6 replicates of ten females. Intestinal permeability was increased upon mating as indicated by the higher number of CFUs in the hemolymph of Oregon-R wild-type mated flies (*p* < 0.05) (**B**) and by the higher level of bromophenol blue dye diffusion in their abdominal area (*p* < 0.001) (**C**). (**D**) *P*. *aeruginosa* infected midguts of mated Oregon-R females had more nuclei marked by the mitosis marker pH3 (*p* < 0.01). (**E**,**F**) Mated 30-days-old females had more *Dl-G4 U-GFP* labelled midgut ISC clusters of 3, 4, 5, 6, 7, 8 or >8 cells each, compared to age-matched virgins (*n* = 12, *p* < 0.001) (**E**), and more clusters of *prospero*-expressing cells (EEs) (*n* = 12, *p* < 0.05) (**F**). (**G**,**H**) Systemic and midgut expression of *attA, diptB, drsl3* and *duox* in the carcasses (**G**) and midguts (**H**), respectively, of mated and virgin Oregon-R females without and upon challenge with heat-killed PA14 (6 replicates of ≥20 female carcasses or midguts). Systemically, *attA* and *diptB* expression was induced upon immune challenge in virgin (*p* < 0.01), rather than mated females (**G**). Midgut expression of *diptB* was reduced in mated females at baseline and upon immune challenge, compared to virgins (*p* < 0.05); *attA* expression was only tentatively decreased in mated females, compared to virgins (**H**). “ns”, “*”, “**”, and “***” stand for *p*-value > 0.05, <0.05, <0.01 and <0.001, respectively.

**Figure 5 metabolites-13-00340-f005:**
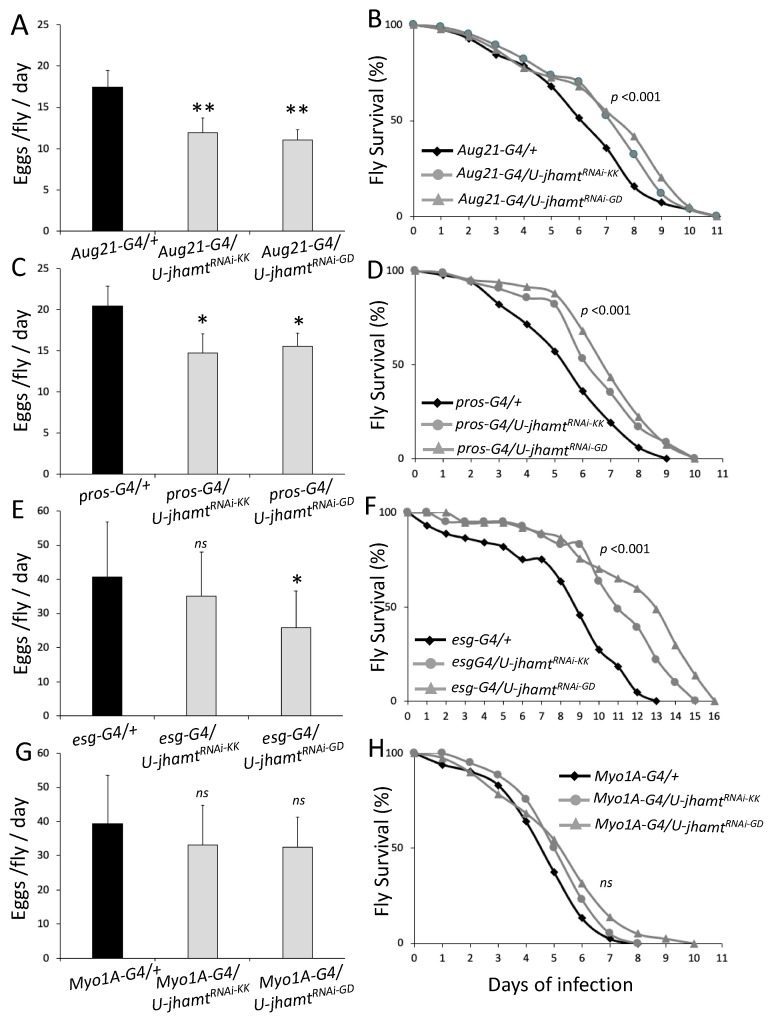
CA and midgut juvenile hormone biosynthesis compromises intestinal host defense. (**A**,**B**) Downregulation of *jhamt* (via VDRC# 103958/KK and 19172/GD) in the CA reduced eggs per fly per day (*p* < 0.01) (**A**), and increased in survival upon infection with *P*. *aeruginosa* (*p* < 0.001) (**B**). (**C**,**D**) Downregulation of *jhamt* in the midgut EEs reduced eggs per fly per day (*p* < 0.05) (**C**), and increased in survival upon infection (*p* < 0.001) (**D**). (**E**,**F**) Downregulation of *jhamt* in the midgut progenitors reduced eggs per fly per day (*p* < 0.05 for 19172/GD) (**E**), and increased in survival upon infection (*p* < 0.001) (**F**). (**G**,**H**) Downregulation of *jhamt* in the midgut enterocytes did not reduce eggs per fly per day (**G**), nor did it increase survival upon infection (**H**). Six replicates of 7–8 flies each were used in (**A**,**C**,**E**,**G**). *n* = 45 flies of each genotype were used in (**B**,**D**,**F**,**H**). “ns”, “*” and “**” stand for *p*-value > 0.05, <0.05, <0.01 and <0.001, respectively.

**Figure 6 metabolites-13-00340-f006:**
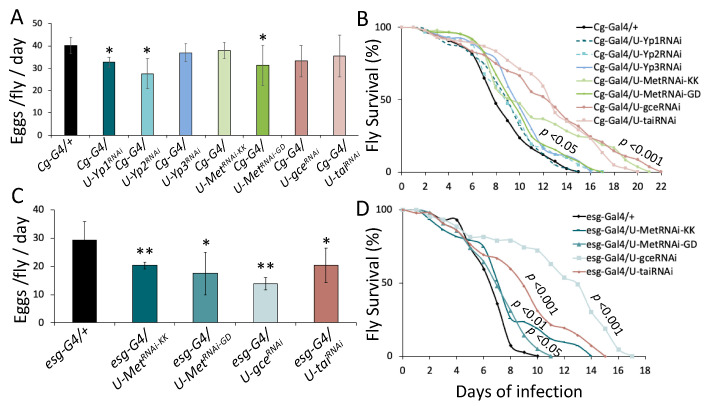
Systemic and midgut progenitor Juvenile Hormone signaling compromises intestinal host defense. (**A**,**B**) Downregulation of *Yp1, Yp2* and *Met* (via VDRC# 10801/GD) in the fat body and hemocytes reduced eggs per fly per day (*p* < 0.05) (**A**) and increased survival upon infection with *P*. *aeruginosa* upon downregulation of *Yp2* (LT50 = 8.85, *p* < 0.05), *Yp3* (LT50 = 9.72, *p* < 0.001), *Met* (100638/KK LT50 = 11.17, *p* < 0.0001; 10801/GD, LT50 = 9.42, *p* < 0.01), *gce* (LT50 = 12.95, *p* < 0.0001) and *tai* (LT50 = 13.65, *p* < 0.0001)), compared to control (LT50 = 7.82) (**B**). (**C**,**D**) Downregulation of *Met, gce* and *tai* in the midgut progenitors reduced eggs per fly per day (*p* < 0.01) (**C**), and increased survival upon infection upon downregulation of *gce* (LT50 = 12.9, *p* < 0.0001), *Met* (100638/KK LT50 = 6.85, *p* < 0.01; 10801/GD lines, LT50 = 7.05, *p* < 0.05) and *tai* (LT50 = 8.75, *p* < 0.0001), relative to control (LT50 = 6.4) (**D**). Six replicates of 7–8 flies each were used in (**A**,**C**); “*” and “**” stand for *p*-value < 0.05 and <0.01, respectively. *n* = 45 flies of each genotype were used in (**B**,**D**).

**Figure 7 metabolites-13-00340-f007:**
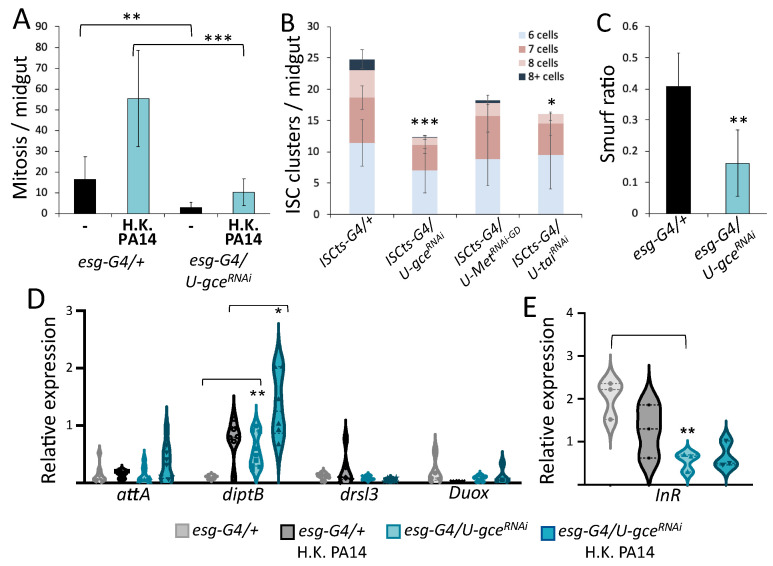
Female midgut progenitor juvenile hormone signaling compromises epithelial integrity and innate immunity and increases InR expression. Downregulation of *gce* in the midgut progenitors reduced mitosis at baseline and upon immune challenge ((**A**), *n* ≥ 10, *p* < 0.01), *ISC^ts^* labelled ISC clusters of 6, 7, 8 and >8 cells ((**B**), *n* ≥ 10, *p* < 0.001), the fraction of Smurf flies ((**C**), *n* = 6, *p* < 0.01), and *InR* expression at baseline ((**E**), 3 replicates of ≥20 midguts, *p* < 0.01), and increased *diptB* at baseline (*p* < 0.01) and upon immune challenge (*p* < 0.05) ((**D**), 6 replicates of ≥20 midguts). “*”, “**”, and “***” stand for *p*-value < 0.05, <0.01 and <0.001, respectively.

**Figure 8 metabolites-13-00340-f008:**
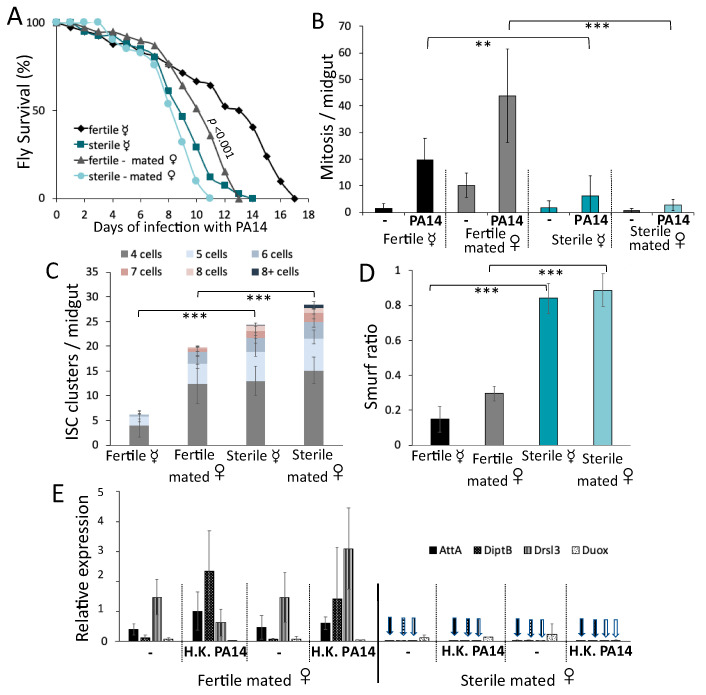
Females compromised in midgut immunity and cell homeostasis are susceptible to intestinal infection, regardless of their virgin vs mated status. (**A**) Virgin and mated sterile *tud^1^/+* females (rendered maternally germline-less) exhibited compromised host defense to intestinal infection, compared to isogenic fertile *tud^1^/+* females (*n* = 30). (**B**–**D**) Virgin and mated sterile *tud^1^/+* females exhibited very low midgut mitosis at baseline and upon infection ((**B**), *n* ≥ 10, *p* < 0.01), and increased intestinal dysplasia ((**C**), *n* ≥ 10, *p* < 0.001) and permeability ((**D**), *n* = 6, *p* < 0.001), compared to isogenic fertile *tud^1^/+* females. (**E**) Mated sterile *tud^1^/+* females exhibited severely reduced midgut expression of *attA, diptB, drsl3* and *Duox* at baseline and upon immune challenge, compared to isogenic fertile *tud^1^/+* females (3 replicates of ≥20 midguts). “**” and “***” stand for *p*-value < 0.01 and <0.001, respectively.

**Figure 9 metabolites-13-00340-f009:**
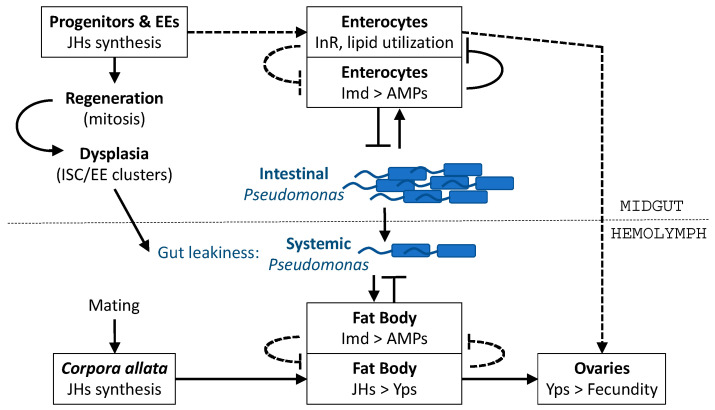
Model diagram of intestinal and systemic interactions between fecundity and intestinal host defence. JHs are synthesized in the midgut progenitors and EEs, facilitating regeneration, but also dysplasia, and in turn leakiness of the gut. Midgut JHs may induce InR signaling and lipid droplet utilization in the enterocytes but may also reduce AMPs and tolerance to intestinal *P*. *aeruginosa*. Vice versa, Imd signaling may reduce InR signaling and lipid droplet utilization in the enterocytes. Mating induces JHs in the CA, and in turn Yps (egg ingredients) in the fat body. Finally, fat body induction of AMPs via Imd signaling may inhibit Yps induction via JH signaling, and vice versa. Arrows and stop arrows indicate actions and counteractions, respectively. Dashed lines indicate hypothetical actions or counteractions.

## Data Availability

All data is contained within the article.

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
