# Peer review of "Intestinal Immune Deficiency and Juvenile Hormone Signaling Mediate a Metabolic Trade-off in Adult *Drosophila* Females"

_metabolites, 2023, doi:10.3390/metabo13030340_

Round 1

Reviewer 1 Report

In this manuscript, Shianiou and Apidianakis investigated the midgut of fly Drosophila melanogaster upon immune challenge, as well as correlating the potential roles of juvenile hormone (JH) in midgut by studying several mutants that are supposed to cause changes in JH levels. Below please find my major comments on the sesquiterpenoid JH section, which hopefully can improve the manuscript.

Major concern/suggestion

1) The authors are investigating an interesting area of how JH signaling can potentially mediate metabolism and fecundity during host defense (Section 2.4). Nevertheless, the authors may have ignored previously published research study (e.g. Wen et al 2017 PLOS Genetics (13(1): e1006559), where authors showed that knockout of JH acid methyl transferase (jhamt) decreased biosynthesis of sesquiterpenoids JH III bisepoxide (JHB3) but not methyl farnesoate (MF), such that the functions of different types of sesquiterpenoids are important (rather than simply juvenile hormone signaling). I strongly suggest the authors would need to measure the JHB3, JH III, and methyl farnesoate (MF) in their different mutants, in order to come up with a solid conclusion and full picture that which sesquiterpenoids maybe functioning in the mid-gut as what the authors proposed. 

I understand that the authors tried to claim support from previous studies (e.g. Rahman et al 2017, Scientific Reports, 7, 11677 - line 463-464), where the authors in that study proposed JHs as key hormonal regulators of gut homeostasis in adult Drosophila. If I understand correctly, the authors in that previous study also didn't measure the titres of different types of sesquiterpenoids. 

Minor/typos

1) line 59 and 60 - delete journal names in the citation (Cell Rep, and PLoS Genetics)

Author Response

Reviewer 1 

“Major concern/suggestion

The authors are investigating an interesting area of how JH signaling can potentially mediate metabolism and fecundity during host defense (Section 2.4). Nevertheless, the authors may have ignored previously published research study (e.g. Wen et al 2017 PLOS Genetics (13(1): e1006559), where authors showed that knockout of JH acid methyl transferase (jhamt) decreased biosynthesis of sesquiterpenoids JH III bisepoxide (JHB3) but not methyl farnesoate (MF), such that the functions of different types of sesquiterpenoids are important (rather than simply juvenile hormone signaling). I strongly suggest the authors would need to measure the JHB3, JH III, and methyl farnesoate (MF) in their different mutants, in order to come up with a solid conclusion and full picture that which sesquiterpenoids maybe functioning in the mid-gut as what the authors proposed. 

I understand that the authors tried to claim support from previous studies (e.g. Rahman et al 2017, Scientific Reports, 7, 11677 - line 463-464), where the authors in that study proposed JHs as key hormonal regulators of gut homeostasis in adult Drosophila. If I understand correctly, the authors in that previous study also didn't measure the titres of different types of sesquiterpenoids.” 

We thank the reviewer for pointing at the existence of three different Juvenile Hormones in Drosophila, namely, JH III, JH bisepoxide (JHB3) and Methyl farnesoate (MF), which can signal through their common JH receptors Met and Gce (Wen et al. 2015). Certainly, JH acid methyl transferase (jhamt) is a key gene in JHs biosynthesis and signaling. Refinement of the literature indicates that Drosophila jhamt controls JHB3 biosynthesis in the corpora allata (CA) and the titer of JHB3 and JH III in the hemolymph during metamorphosis (Wen et al. 2015). It is thus clear that the downregulation of jhamt in the Drosophila CA and the midgut cells, where it is evidently expressed (Rahman et al 2017; Dutta et al 2015), is a valid approach to reduce the systemic and the local level, respectively, of one or more of the Drosophila JHs and JH signaling through the JH receptors. We now provide this information in the introduction (lines 56-65). Moreover, we discuss that further work may focus on the biochemical measurement of the three JHs in genetically manipulated adults to clarify the chemical identity of JHs produced by specific enzymes in specific cell types and tissues, such as the midgut enteroendocrine and progenitor cells, and the CA of Drosophila adults. Biochemical analysis may, nevertheless, be complex, because enteroendocrine or progenitor cells of the midgut may differ from each other and from CA cells in the relative abundance of the JHs they produce. Moreover, midgut enteroendocrine or progenitor cells may act mainly in a paracrine or endocrine way and contribute only a small fraction of the total JHs produced in the body (lines 525-529). Thus, their systemic contribution may be limited and undetectable in the hemolymph. We find that such experiments are important but lie beyond the approach of this work.

Wen D, Rivera-Perez C, Abdou M, Jia Q, He Q, Liu X, et al. (2015) Methyl Farnesoate Plays a Dual Role in Regulating Drosophila Metamorphosis. PLoS Genet 11(3): e1005038

Rahman, M.M., Franch-Marro, X., Maestro, J.L., Martin, D., Casali, A. (2017). Local Juvenile Hormone activity regulates gut homeostasis and tumor growth in adult Drosophila.  Sci. Rep. 7(1): 11677

Dutta D, Dobson AJ, Houtz PL, Gläßer C, Revah J, Korzelius J, Patel PH, Edgar BA, Buchon N. Regional Cell-Specific Transcriptome Mapping Reveals Regulatory Complexity in the Adult Drosophila Midgut. Cell Rep. 2015 Jul 14;12(2):346-58

“Minor/typos

line 59 and 60 - delete journal names in the citation (Cell Rep, and PLoS Genetics)

A formatting of citations was requested by the journal editors, so in the current version of our manuscript all reference citations are shown as mere numbers.

Reviewer 2 Report

The manuscript by Shianiou and Apidianakis describes a complex study devoted to the check of hypothesis of a trade-off between reproduction and survival to infection on Drosophila melanogaster model. The authors succeeded obtaining evidence supporting this hypothesis and demonstrated such a trade-off taking place in Drosophila females. The results obtained in the study provide new insights into the molecular mechanisms underlying antagonism between intestinal defense and fecundity. The study is well setup, looks well executed and contains new data which are well discussed. The manuscript could be of interest to the readership of Metabolites.

However, I see two points in the Introduction of manuscript that have to be improved.

1)     It is better to add some information concerning genes which were taken into analysis in this study. It is rather confusing when they appeared out of the blue in the Results.

2)     Authors mentioned Corpora allata in relation to Drosophila. But it has only one Corpus allatum, not two! Authors refered to the paper by Toivonen and Partridge (2009) where this could be seen on very clear scheme.

Minor comments:

20-hydroxyecdysone usually is written without hyphen between the parts of the word.

Author Response

Reviewer 2

“The study is well setup, looks well executed and contains new data which are well discussed. The manuscript could be of interest to the readership of Metabolites.”

We thank the reviewer for appreciating the overall significance of our work.

“However, I see two points in the Introduction of manuscript that have to be improved.

1)     It is better to add some information concerning genes which were taken into analysis in this study. It is rather confusing when they appeared out of the blue in the Results.”

We have now added a thorough introduction on a ket gene in our analysis, jhmpt (lines 61-65). We also mention the primary function of CG11307 and Ms in gut motility in lines 81-82, including the pertinent citation.

“2)     Authors mentioned Corpora allata in relation to Drosophila. But it has only one Corpus allatum, not two! Authors refered to the paper by Toivonen and Partridge (2009) where this could be seen on very clear scheme.”

We have now replaced throughout our manuscript “corpora allata” with “corpus allatum” or “CA”, as correctly indicated by the reviewer.

“Minor comments:

20-hydroxyecdysone usually is written without hyphen between the parts of the word.”

We have now replaced “20-hydroxy-ecdysone” with “20-hydroxyecdysone”, as correctly indicated by the reviewer.

Reviewer 3 Report

In this study, Gavriella Shianiou and Yiorgos Apidianakis explored the trade-off between fecundity and host defense by using the Drosophila model, and they found a mutual antagonism specifically between Imd immune signaling in the midgut enterocytes and JH-mediated fecundity. Overall, this is an interesting study that contributes to the knowledge of how animals optimize the allocation of limited resources to multiple energy demanding processes, including reproduction and host defense. The conclusions obtained are convincing and well-supported to some degree. However, I still have some concerns that should be specified or clarified.

1) Authors need to check the defecation performance of flies after moto/Ms-knockdown, to pinpoint the link between intestinal motility and fecundity, or use other RNAi lines of crucial genes in defecation shown previously (like TrpA1) to verify that gut motility/defecation can impact fecundity.

2) The expression of myosuppressin is super low in midgut, it is mainly expressed in the brain (PMID: 18972134). Using Pros-Gal4 driver to repress MS expression in EEs does not make sense.

3) Did authors check the related phenotypes after infection in male flies?

4) Heat-killed bacteria should be nonpathogenic, but still can decrease egg laying like live pathogenic bactria. Do heat-killed bacteria exert immune burden to flies? Survival assay needs to be conducted. Moreover, authors also need to discuss related mechanisms in discussion.

5) Virgin males and mated males should be considered as negative controls when getting the conclusion that mating compromises intestinal host defense.

6) Besides survival assay, authors should use more specific tools to assess the intestinal host defense.

7) More details should be provided about the survival assay, including how many flies were used, and source of P. aeruginosa strain PA14.

8) It would be great to put a model diagram in the last figure to summarize the story.

9) Double check the text and format to avoid careless mistakes, eg.: Line 85, “(KK108230”, multiple fonts were used in the manuscript.

Round 2

Reviewer 1 Report

I accept the authors' explanation, and so this manuscript can now be published in this journal.

Reviewer 3 Report

I am now satistified with this manuscript after these modifications.